# Identification of Transferrin Receptor 1 (TfR1) Overexpressed in Lung Cancer Cells, and Internalization of Magnetic Au-CoFe_2_O_4_ Core-Shell Nanoparticles Functionalized with Its Ligand in a Cellular Model of Small Cell Lung Cancer (SCLC)

**DOI:** 10.3390/pharmaceutics14081715

**Published:** 2022-08-17

**Authors:** Rocío Villalobos-Manzo, Emmanuel Ríos-Castro, José Manuel Hernández-Hernández, Goldie Oza, Mauricio A. Medina, José Tapia-Ramírez

**Affiliations:** 1Departamento de Genética y Biología Molecular, Cinvestav-IPN, Ciudad de México 07360, Mexico; 2Unidad de Genómica, Proteómica y Metabolómica (UGPM), LaNSE, Cinvestav-IPN, Ciudad de México 07360, Mexico; 3Departamento de Biología Celular, Cinvestav-IPN, Ciudad de México 07360, Mexico; 4Centro de Investigación y Desarrollo Tecnológico en Electroquímica (CIDETEQ), Parque Tecnológico de Querétaro s/n Sanfandila, Pedro Escobedo, Querétaro 76703, Mexico; 5Programa de Nanociencias y Nanotecnología, Cinvestav-IPN, Ciudad de México 07360, Mexico

**Keywords:** label-free, mass spectrometry, nanoparticles, small cell lung cancer (SCLC), transferrin receptor (TfR1), cluster of differentiation 71 (CD71)

## Abstract

Lung cancer is, currently, one of the main malignancies causing deaths worldwide. To date, early prognostic and diagnostic markers for small cell lung cancer (SCLC) have not been systematically and clearly identified, so most patients receive standard treatment. In the present study, we combine quantitative proteomics studies and the use of magnetic core-shell nanoparticles (mCSNP’s), first to identify a marker for lung cancer, and second to functionalize the nanoparticles and their possible application for early and timely diagnosis of this and other types of cancer. In the present study, we used label-free mass spectrometry in combination with an ion-mobility approach to identify 220 proteins with increased abundance in small cell lung cancer (SCLC) cell lines. Our attention was focused on cell receptors for their potential application as mCSNP’s targets; in this work, we report the overexpression of Transferrin Receptor (TfR1) protein, also known as Cluster of Differentiation 71 (CD71) up to a 30-fold increase with respect to the control cell. The kinetics of endocytosis, evaluated by a flow cytometry methodology based on fluorescence quantification, demonstrated that receptors were properly activated with the transferrin supported on the magnetic core-shell nanoparticles. Our results are important in obtaining essential information for monitoring the disease and/or choosing better treatments, and this finding will pave the way for future synthesis of nanoparticles including chemotherapeutic drugs for lung cancer treatments.

## 1. Introduction

Lung Cancer is the most frequent cause of lethality among all types of cancer worldwide with an estimated 2.2 million new cases of lung cancer per year, and 1.79 million deaths by the year 2020. (International Agency for Research on Cancer) [1]. There is strong evidence linking smoking to 85% of the incidence of lung cancer [2], as well as further risk factors such as occupational exposure to asbestos and combustion fumes, and environmental exposure to arsenic and air pollution. Whilst these factors remain the major contributors in the developing world, 15% of lung cancer cases occur in the lives of those who were never smokers [3]. Inherited genetic factors are thought to play a minor role in the susceptibility to developing SCLC. Genetic variation does contribute to the risk of nicotine addiction and might thereby indirectly influence SCLC risk [4].

The main challenge in improving the poor survival rate (5-year survival, approximately 15%) of this disease is to develop better strategies to stratify high-risk populations, for early diagnosis and selection of appropriate treatment for different subsets of lung cancer. The mortality associated with this disease is high, primarily because most lung cancers are diagnosed at advanced stages when the options for treatment are mostly palliative [5].

Lung tumors are the result of a multistep process in which normal lung cells accumulate multiple genetic and epigenetic abnormalities and progress into cells with malignant biologic capabilities [6]. The two main types of lung cancer, identified based on the histologic, clinical, and neuroendocrine characteristics, are small cell lung cancer (SCLC, representing 15% of the cases) and Non-small cell lung cancer (NSCLC, representing approximately 85% of the cases). Although NSCLC displays numerous histologic patterns, most tumors can be grouped into three main categories: squamous cell carcinoma (30%), adenocarcinoma (40%), and large cell carcinoma (3–9%) [7]. NSCLC and SCLC differ molecularly, showing many genetic alterations and exhibiting subtype specificity [8].

SCLC is a poorly differentiated high-grade neuroendocrine tumor that is commonly located centrally in the major airways but may occur peripherally in the lungs in about 5% of the cases. SCLCs are typically situated in a peribronchial location with infiltration of the bronchial submucosa and peribronchial tissue [5] presenting a high growth rate and early development of metastases and have a strong association with cigarette smoking. Incidence rates of SCLC are higher in men than in women, but a higher percentage of lung cancers are of the SCLC type in women than in men [9]. SCLC consists of cytologically malignant epithelioid cells with scant cytoplasm, nuclei with granular (“salt and pepper”) chromatin without prominent nucleoli, and more than 10 mitoses per 2 mm^2^ (usually more than 50 mitoses per 2 mm^2^) [10]. The tumor cells are arranged in sheets, but there are also rosettes, trabeculae, or peripheral palisading of cells along the edges of nests [5].

Progress in the treatment of SCLC has been very limited. The traditional treatments used to deal with this type of pathologies are radiotherapies, chemotherapies, and surgeries. Platinum-etoposide-based regimens have been the mainstay of treating this disease for many years. There have been very few signs of success in targeted therapy, and there are none in routine clinical use [11]. Moreover, in most patients, the therapy causes side effects, which lead to a low quality of life [12].

The treatment of patients with SCLC is complicated, since they usually have multiple important comorbidities due to tobacco use such as chronic obstructive pulmonary disease, ischemic cardiopathy, and hypertension, thus deteriorating their functional status. In addition, SCLC is highly aggressive and is accompanied in general by significant weight loss, fatigue, and symptoms related to bulky intrathoracic disease and/or metastasis that contribute to the patients’ frailty and obstruct optimal oncologic treatment [13].

In this scenario, the academy and pharmaceutical industry conduct their research to propose alternative diagnostic methods and treatments. In this sense, nanomedicine seems to be a promising technology to diagnose and also treat lung cancer in general. The designing bases of cancer nanomedicine, in particular for solid tumors, are: (i) the enhanced permeation and retention effect (EPR) and (ii) the cancer cell-specific affinity targeting. Their appropriate design, as well as the study of their impact on cancer nanomedicine, extends the therapeutic window by enhancing the efficacy and reducing the toxicity [14,15]. For instance, more accumulation in target sites, thereby less exposure to other healthy cells, is expected by the identification of the overexpressed membrane receptors on the cancer cells.

Several research groups have been working with magnetic and plasmonic nanoplatforms and have been evaluating their behavior as imaging contrast agents, hyperthermia properties, and obviously their application in the controlled delivery of drugs, however, the specificity of recognition of the nanoplatforms has been very limited.

In the present study, we used label-free mass spectrometry in combination with an ion-mobility (IM-MS) approach to evaluate differences in the proteome of enriched membranes in a human aggressive and multidrug-resistant SCLC cell line H69AR (ATCC^®^ CRL-11351). This cell line was used to generate tumors in immunodeficient mice to carry out an in vivo analysis (data not shown); also, MRC5 (ATCC^®^ CCL171), a non-tumor fibroblast lung cell line was used. In this way, we were searching for which protein receptors are up-regulated in cancer versus normal cells. One of the most up-regulated membrane receptors in SCLC cell lines was identified as Transferrin Receptor 1, TfR1. Together, the finding and the experimental results allowed us to propose transferrin as a ligand on magnetic core-shell nanoparticles (mCSNP’s) because it leads to proper nanoparticle endocytosis through a receptor. The identification of the TfR1 receptor, overexpressed in lung cancer, will be essential to treat this and other types of cancer in the near future.

## 2. Materials and Methods

### 2.1. Samples

SCLC cell line H69AR (ATCC^®^ CRL-11351), used as cancer cell line, was cultured in an RPMI-1640 medium supplemented with 10% fetal bovine serum (FBS), L-glutamine (2 mM), and antibiotics (100 U/mL penicillin and 100 U/mL streptomycin). A lung fibroblast cell line MRC5 (ATCC^®^ CCL171) was used as the control. The latter cell line was cultured in a Dulbecco’s Modified Eagle’s Medium supplemented with 10% FBS and antibiotics in a 5% CO_2_ humidified incubator at 37 °C.

### 2.2. Sample Processing for Mass Spectrometry

The identification of the overexpressed receptor SCLC accounted for the extraction. The membrane protein extraction of both cell lines was performed with a Mem-PER™ Plus Membrane Protein Extraction Kit (Thermo Scientific™, Waltham, MA, USA), after collecting the proteins, a protease inhibitor cocktail was added to maintain the integrity of the proteins; equal amounts of protein were prepared for proteomic studies. The kit allows preferential solubilization of integral membrane proteins and membrane-associated proteins, the degree of contamination with cytosolic proteins is less than 10%.

For each biological sample (H69AR and MRC5 cell lines), 6 × 10^6^ cells were harvested, washed, and centrifuged. The cytosolic proteins were isolated in the supernatant fraction after adding Permeabilization Buffer™ and mixing at 4 °C for 10 min. The homogenate was centrifuged at 16,000× *g* at 4 °C for 10 min; membrane proteins were isolated in the supernatant fraction after adding Solubilization Buffer™ and mixing at 4 °C for 30 min; the homogenate was centrifuged at 16,000× *g* at 4 °C for 15 min. Protein concentrations were measured using a 2-D Quant Kit™ (GE Healthcare™, Chicago, IL, USA) according to the manufacturer’s recommendations. Protein pool (40 μg) for each condition and biological replicate were subjected to 10% SDS-PAGE and allowed to advance for about 1 cm within the gel; the resulting gel fragments were enzymatically digested according to the modified protocol of Shevchenko et al. 2006 [16].

### 2.3. Relative Quantification by Label-Free DIA Mass Spectrometry

The quantitative proteomic analysis (spectrometric and chromatographic conditions) was performed in a UPLC Nano Acquity *M*-Class coupled with a QTOF Mass Spectrometer Synapt G2-S*i* (Waters; Milford, MA, USA), according to the method of Ríos-Castro et al., 2020 [17].

### 2.4. Data Analysis

Generated *.raw files were analyzed in Progenesis QI for Proteomics *v*3.0.3 software (Waters; Milford, MA, USA) according to the settings reported in the study of Vázquez-Procopio J, et al., 2020 [18], with the following modifications: we used a concatenated *.fasta *Homo sapiens* database (downloaded from Uniprot, containing 73,099 protein sequences, last modification on 27 June 2018). Synapt G2-S*i* was calibrated with [Glu1]-fibrinopeptide fragments through the precursor ion [M + 2H]^2+^ = 785.84261 fragmentation of 32 eV with a result of less than 1.1 ppm across all MS/MS measurements. The ratio was calculated based on the average MS signal response of the three most intense tryptic peptides (Top3) of each characterized protein in the H69AR (tumor cells) sample by the Top3 of each characterized protein in the MRC5 (normal cells) sample. The mass spectrometry proteomics data have been deposited to the ProteomeXchange Consortium via the PRIDE [19] partner repository with the dataset identifier PXD015405.

### 2.5. Bioinformatic Analysis

Expression profiles were visualized as a heat map, which was generated using a web server heatmapper [20], available online (http://heatmapper.ca/; accessed on 28 February 2020). Complete linkage and Pearson methods were used as clustering and distance measurement methods, respectively. “*z*-Score” was calculated with the following equation:$$z=\frac{\;x-\mu }{\sigma }\;$$
where, x = Top3 abundance of a protein in a single injection, *µ* = mean abundance Top3 per protein (all injections), and *σ* = standard deviation.

Differentially expressed proteins were classified according to the biological pathways using the Reactome database contained in STRING *v*11 (available online: https://string-db.org/; accessed on 8 November 2019). All resulting biological pathways with an FDR (False Discovery Rate) ≤ 0.05 (highly reliable) were exported as a *.tsv file to be plotted (FDRs were converted to −log_10_) as a bar graph using TIBCO Spotfire software *v*7.0.0 (Somerville, MA, USA). Interactomes of differentially expressed proteins were made in STRING too, with the following settings: *Homo sapiens* database; textmining, experiments, database, co-expression, neighborhood, gene fusion, and co-occurrence as active interaction source, and 0.4 as a confidence score. A TfR1 interactome was constructed using Cytoscape software *v*3.8.0 (available for download: https://cytoscape.org/) through the ReactomeFIPlugIn *v*7.2.3 application.

### 2.6. Western Blot

Based on spectrometric analysis results, western blot analysis was performed to confirm the overexpressed receptor in SCLC. Equal amounts (20 μg) of protein were prepared and run in SDS/PAGE gel; proteins were transferred onto a nitrocellulose membrane. The membrane was incubated overnight with goat anti-CD71 (Santa Cruz Biotechnology, sc-32272, Dallas, TX, USA) and monoclonal anti-β-actin at 4 °C. After washing, membranes were incubated with horseradish peroxidase-conjugated secondary antibody for 2 h at room temperature. The bands containing the proteins were visualized on an X-ray film (Kodak, Rochester, NY, USA) using an enhanced chemiluminescence (Western Lightning Plus-ECL, PerkinElmer, Inc. Waltham, MA, USA) kit. Densitometric analysis of Western blot bands was performed using the software Image StudioTM Version 5.2.5. (Lincoln, NE, USA).

### 2.7. Flow Cytometry

#### 2.7.1. For the Overexpressed Receptor

The monolayer H69AR (ATCC^®^ CRL-11351) and MRC5 (ATCC^®^ CCL171) cells were washed with PBS and incubated with trypsin-EDTA at 37 °C for 5 min. The cells were collected and re-suspended in a medium. Survival of all cells was counted using a trypan blue dye exclusion assay. The cell number in the suspension was adjusted to 5 × 10^5^ cells per sample per cell line (sample 1: Blanc and sample 2: mix of folate receptor and CD71 receptor antibodies). The cells were washed twice with 1 mL of PBS to remove the culture medium by centrifugation for 5 min. Cells were resuspended in 100 μL of PBS-EDTA and then sample 1 was incubated, without specific antibody, only with 0.5% FBS, and sample 2 was incubated with anti-Human FOLR1 AF488-conjugated antibody (R&D Systems, FAB5646G, Minneapolis, MN, USA) and mouse anti-Human CD71 PE-conjugated antibody (BD Pharmingen, 555537, East Rutherford, NJ, USA). After incubating for 30 min, the samples were analyzed immediately by flow cytometry.

#### 2.7.2. For the Endocytosis Studies

The cell number in the suspension was adjusted, and 1.5 × 10^5^ cells in 1000 μL of medium were seeded to use for the cellular uptake test. After incubating the samples with 2 μg/mL Transferrin Alexa Fluor™ 488 Conjugate (Tf-AF488, Invitrogen Molecular Probes, Carlsbad, CA, USA) and transferrin conjugated gold nanoparticles for 0.5, 8, and 16 h at 37 °C in a humidified incubator (5% CO_2_), the cells were washed twice with 3 mL of PBS to remove untaken transferrin by centrifugation for 5 min. The cell suspensions were suspended in 100 μL of PBS-EDTA and analyzed immediately by flow cytometry.

We added 0.2 μg of the TF-AF488 conjugate to have positive control of the endocytic level in both cell lines (MRC5 normal and H69 cancer cell lines). Total TF-AF488 fluorescence at *t* = 0.5, 8, and 16 h was measured by flow cytometry as a positive-maximum endocytic signal. Regarding the CSNP–TF complex, we added a respective volume to get the TF final concentration of 0.2 μg in the medium of incubation.

### 2.8. Synthesis of Nanoparticles

#### 2.8.1. Synthesis of Monodispersed Cobalt Ferrite Nanoparticles

The synthesis and characterization of monodisperse cobalt ferrite nanoparticles (CFNPs) have been previously described [21]. Cobalt acetylacetonate and Iron acetylacetonate react with the nonionic surfactant Triton-X100, the synthesis of nanoparticles of Fe_2_O_4_ magnetite doped with cobalt was carried out by the process of Thermal Decomposition (TD), which involves the reaction of a precursor compound that occurs by increasing the temperature (roughly 250–270 °C) to produce two or more compounds. *TD* of iron(III) acetylacetonate 1 mmol (Sigma 97%) precursor plus cobalt acetylacetonate 0.5 mmol (Sigma Aldrich 97%, San Louis, MO, USA) plus an organic surfactant as Triton X-100 (0.1 M).

The mixture was placed in a volumetric flask connected to a cooling system driven by ice water and a pump to circulate the cold water inside a refrigerant tube and allow the reaction to be carried out for the required time under Nitrogen (N_2_) atmosphere, during 10 min of exposure and a temperature of 260 °C, the solution was heated for 60 min until it acquires the characteristic dark color of magnetite.

Approximately 15 mL solution was left at room temperature for 24 h and subsequently centrifuged in 5 cycles at 8000 RPM, the first three adding 5 mL of methanol 98% and the last two cycles adding 5 mL water milli Q 800 Ohm.

The characterization showed the presence of CoFe_2_O_4_ nanoparticles of 5 nm on average with superparamagnetic properties. The morphology and size of cobalt ferrite nanoparticle were characterized using transmission electron microscopy (TEM JEM-ARM200F, Jeol, Tokyo, Japan) and HAADF-STEM (High-Angle Annular Dark-Field scanning Transmission Electron Microscopy). In order to reduce the toxicity of the cobalt-iron nanoparticle, a gold coating was formed on the nanoparticle. The coating is fundamental to get a uniform sample and enhanced biological capacities, otherwise, CoFe_2_O_4_ would be toxic for “In Vitro’’ or animal models. The gold coating also drives for an enhanced half-life in the system.

#### 2.8.2. Synthesis of Magnetic Au-CoFe_2_O_4_ Core-Shell Nanoparticles (mCSNP’s)

The synthesis of Au seed solution (Solution 1) was prepared by using 0.5 mL (1 M, CTAB), 1 mL (50 mM, L-Ascorbic acid), and 100 µL (1 M, HAuCl_4_ solution). The whole complex was sonicated for 30 min. The freshly prepared gold seed solution was used for Au shell coating. The preparation of cobalt ferrite nanoparticles was used as solution 2. Solutions 1 y 2 were used to synthesize the core-shell. Initially, 1 part of solution 2 was added dropwise to solution 1. This solution mixture was stirred for 3 h continuously until there was a color change. Then, these core-shell nanoparticles were magnetically separated using a magnet and they were washed twice with a mixture of hexane and ethanol to obtain high purity NPs excluding excess gold NPs. The washed NPs were centrifuged again twice to remove excess Triton X-100 from the solution.

### 2.9. Functionalization of Au Core-Shell Magnetic Nanoparticles

Modification of the gold core-shell was carried out using 11-mercaptoundecanoic acid (MUA). Subsequently, transferrin (Transferrin -Alexa Fluor 488 of Invitrogen) was conjugated to the core-shell; the transferrin-functionalized nanoparticles were separated from the free transferrin by washing and concentration with a magnet. The methodology (Figure 1) was adapted with some modifications from Springer Protocols [22]. After the functionalization process, electrophoresis (Agilent 2100 Bioanalyzer, Santa Clara, CA, USA) was carried out to determine if the transferrin had been properly bound to the nanoparticle, for which the supernatant containing transferrin dissolved in PBS was analyzed. 

### 2.10. Immunofluorescence of TfR1 in Cells

The cells MRC5 and H69AR were seeded on coverslips, after 24 h they were washed three times with PBS 1X after they were fixed with 4% (*w*/*v*) paraformaldehyde in PBS for 30 min. The cells were not permeabilized because the location of the protein receptor of our interest is in the membrane. The anti CD71 ab diluted 1:500 was incubated at 4 °C overnight; the secondary goat anti-mouse-AF488, 1:750 was incubated, and the samples were embedded using Vectashield HardSet Mounting Media (Vector Laboratories, Burlingame, CA, USA). Subsequently, the processed samples were analyzed using an epifluorescence confocal laser scanning microscope Cytation C10 (BioTek, Santa Clara, CA, USA).

## 3. Results

### 3.1. Proteomic Analysis

As a result of quantitative proteomics analysis; 123,724 peptides were detected (ions *z* = 2^+^ or superior) in the entire study, and 86.92% of these peptides fell at a maximum of ±10 ppm. A complete analysis at the peptide level exhibited an adequate adjustment in terms of calibration, ionization source operability, and enzymatic effectiveness (Appendix A). These peptides represent 1298 quantified proteins shared in both cell lines with an average of nine peptides per protein in a dynamic range of ~6.5 orders of magnitude (expressed as log10 logarithm) revealing good sensitivity of the spectrometer and correct normalization of the injection since both dynamic ranges (MRC5 and H69AR) are properly adjusted (Appendix A). Quantified proteins were filtered (CV ≤ 0.30, at least two peptides/protein, considering at least one unique peptide, proteins that replicated only 3/3, and ANOVA *p*-value ≤ 0.05); additionally, “reversed” proteins were discarded. Through the expression analysis performed with a heat map, the formation of four quadrants differentiated from each other in terms of abundance could be visualized, indicating marked differences in the expression profile of membrane proteins between both cell lines (Appendix A), what was expected because of the nature of both cell lines [23,24], however, not all the proteins analyzed in the heat map were considered as differentially expressed. For such reason, we scattered the proteins in a volcano plot using a cut-off value of 1.2, expressed as base 2 logarithm (log_2_) [4,17], and finally, 188 proteins up-regulated in H69AR cells and 174 in MRC5 cells were reported (Appendix A); besides, we reported 32 and 33 exclusive proteins in H69AR and MRC5, respectively (Appendix A). All proteins identified in this work with their corresponding spectrometric measurements are summarized in Appendix A.

### 3.2. Bioinformatics

Differentially expressed proteins (427 proteins) were analyzed in STRING and the results show that the proteins participate in 212 biological pathways according to Reactome (Appendix A). Additionally, within the top 50 biological pathways (Figure 1), some of them, like the metabolism of RNA (FDR = 2.06 × 10^−17^), the formation of a translation initiation complex (FDR = 1.94 × 10^−14^), and the cell cycle pathway (FDR = 8.02 × 10^−5^), actively participate in processes related to the development of tumors such as those involved in the cell cycle, immune response, protein synthesis, and vesicle transport.

Visualizing the interactome of the differentially expressed proteins, we could notice a high degree of interconnection between them (Appendix A), which hints at totally differentiated molecular mechanisms between cell lines. Many of the most interconnected proteins are those involved with vesicular trafficking; but among them, we focus our attention on TfR1 (TFRC, CD71) because it impacts the function of many other proteins of vesicle trafficking such as ACTR3, AP2A1, ACTR2, ARPC3, YWHAZ, and even HIP1, which regulates the assembly of clathrin cages [25,26], through its direct interaction [27]; consequently, these proteins trigger effects on downstream pathways like those involved in the cell cycle, immune response, and RNA metabolism and translation (Figure 2). The analysis of the biological pathways of the interactome, in Figure 2, allows us to observe several protein clusters that are important from the point of view of carcinogenic processes, for example for DEAD-Box RNA helicases; the cluster of nuclear heterogeneous Ribonucleoproteins (HNRNPA1, HNRNPA2B1, and other), which are involved in DNA repair processes, chromatin remodeling and regulation of gene expression, or the cluster of translation initiation factors that have been found to be overregulated in several types of cancer (Figure 2). 

Metabolism of RNA was the main biological pathway altered, in which several clusters of proteins showed participation, such as DEAD-box RNA helicases (DDX21, DDX39A, DDX39B, DDX46, DDX5, and DHX9). These proteins are important in the carcinogenic process because many of them are involved in the progression and arrest of the cell cycle, cell migration and invasion, apoptosis, coactivation of transcription factors involved in tumor development, and aberrant regulation of RNA, which triggers cell growth [28].

Another important cluster is formed with heterogeneous nuclear ribonucleoproteins (HNRNPA1, HNRNPA2B1, HNRNPC, HNRNPD, HNRNPF, HNRNPH2, HNRNPK, HNRNPR, HNRNPU, and HNRNPUL1), which participate in a wide range of roles in DNA repair, telomere biogenesis, chromosome remodeling, cell signaling, and regulating the gene expression at both the transcriptional and translational levels by direct influence on pre-mRNA splicing through site-specific binding with the target RNA [29,30].

The biological pathway involved in the formation of a translation initiation complex is also well represented because it is formed by other clusters containing eukaryotic translation initiation factors (EIF3B, EIF3K, EIF4A1, EIF4A2, and EIF4H). The eIF3 subunits in conjunction with other eIF subunits help to stabilize the 40S ribosomal subunit and many of them are up-regulated in different types of cancer such as breast, cervix, lung, squamous cell, colorectal, neuroblastoma, prostate, and non-small cell lung cancers. On the other hand, other subunits like eIF3F have been reported to be down-regulated in breast cancer, vulvar cancer, pancreatic cancer, and ovarian cancer among others [31]. 

In this work, all eukaryotic translation initiation factors are reported to be up-regulated and in agreement with this result, and we detected several subunits of ribosomal proteins (RPS10, RPS11, RPS12, RPS14, RPS16, RPS18, RPS19, RPS23, RPS24, RPS3, RPS4X, RPS6, RPS8, and RPSA) up-regulated too, indicating a high presence of translational events. This dysregulation is often associated with aberrant function and perturbations in the expression of components of the translation machinery in cancer [32,33].

The presence of proteins such as hNRP, DEAD-Box RNA helicases, eukaryotic translation initiation factors, or ribosomal proteins, which a priori could not be considered as membrane proteins, is constant in different proteomic studies of membrane enrichment in different cell lines. In a recent article Statello et al. 2021 [34], mention that the interaction of several HNRNPs with RNA facilitates the transport of RNAs into exosomes. One possibility of finding these HNRNPs in our analysis is that in the extraction process, we are including the exosomes and the proteins included within these extracellular vesicles. On the other hand, exosomes are matured into multivesicular bodies from late endosomes [35,36] and can carry proteins associated with the endoplasmic reticulum or proteins that are in the process of translocation, elsewhere, in the cell. In cancer cells, the vesicles or exosomes are found to be loaded with different types of biomolecules (DNA, non-coding RNAs, miRNA, tumor antigens, proteins) that do not appear in non-cancerous cells.

### 3.3. Validation of Differential Expression of TfR1

Differential expression of TfR1 using MS showed marked differences between the two cell types. To put this in context, TfR1 is the 27th protein in the dynamic range of the H69AR cell line proteome, while the MRC5 dynamic range is 725th (Figure 3A); additionally and considering only up-regulated proteins in the tumor cell line, TfR1 is 15th in terms of abundance and only one receptor was observed as more abundant, the membrane-associated progesterone receptor component 1 (PGRMC1), but was not selected because its relative abundance in H69AR is 9.33 fold compared to 32.86 fold of TfR1 in MRC5 and because PGRMC1 is less selective, it can bind a wide variety of molecules like sterols, hemes, and progesterone [37].

Parallel to MS, the differential expression of TfR1 was corroborated using other methodologies like Western blot; we observed TfR1 band was more expressed in the cancer cell protein sample than in the normal cell sample (Figure 3(B.1)), this result is in line with the literature reporting that TfR1/Cd71 (a homodimeric type II membrane glycoprotein, ∼95 kDa) binds to, and assists entry of its ligand into cells for the delivery of iron, the plot evidencing increased abundance (Figure 3(B.2)), and flow cytometry (Figure 3C) in which we observed that fluorescence in the plot is 90% (Figure 3(C.3)) in quadrant B−+ corresponding to Tf (Figure 3(C.2)); both techniques showed much higher expression in the H69AR cell line compared to the MRC5 cell line, noting that the result of three different techniques was concordant.

### 3.4. Immunofluorescence of Transferrin Receptor

The immunofluorescence technique was used as another way to corroborate the presence of the protein in the plasmatic membrane of the tumoral cells and its overexpression on them. As we can observe in Figure 4 the intense green signal corresponds to the TfR1. 

### 3.5. Synthesis and Functionalization of mSCNP’s

In the present work, we use the thermal decomposition method, to carry out the synthesis of cobalt-iron nanoparticles with magnetic properties, characterization, and hyperthermia properties that have been previously reported [21,38]. In Figure 5 we show the results of transmission electron microscopy when analyzing a preparation of the magnetic cobalt-iron nanoparticles with the gold shell, performed under the conditions described in material and methods, the morphology and size of the magnetic Au-CoFe_2_O_4_ core-shell nanoparticles (mCSNPs) were homogeneous, nearly monodisperse, spherical and of a regular size of 5–10 nm. Further, high-angle annular dark-field imaging was obtained from preparation of Au-CoFe_2_O_4_ Core-shell, the scan mapping revealing a core-shell nanoparticle with Au shell and Co and Fe as a core (Figure 6). 

In the second part of this work, we carried out the functionalization of our core-shell nanoparticle (Au-CoFe_2_O_4_) with the TfR1 target molecule, the transferrin (Tf). The objective was to increase the efficiency with which the core-shell nanoparticle can recognize the target cancer cell. The efficiency with which transferrin binds to the Au-CoFe_2_O_4_ core-shell nanoparticle depends, among other factors, on the length of the linker and the protein, in our case we used an 11-carbon linker (MUA). The binding of transferrin was monitored by microfluidic electrophoresis on the Agilent 2100 system. The transferrin bound to the nanoparticle can be separated by a magnet, and the unbound transferrin remains in the supernatant. The initial concentration in the binding assay was 15 µg of Tf. The gel image (Appendix A), shows the 79 kDa band corresponding to transferrin glycoprotein. When quantitative analysis of this band was performed, the amount of protein obtained in the supernatant was 10.34 µg of protein, therefore the difference of 4.66 µg corresponds to the core-shell nanoparticle-transferrin complex. With this complex, we proceeded to perform the transferrin receptor-mediated endocytosis analysis.

### 3.6. Endocytosis Assays

Finally, the endocytosis assay for transferrin was performed allowing the activation of TfR1 according to the results of flow cytometry related to the endocytosis of the mCSNP–TF complex (Figure 7). Figure 7(a.1) shows the blank lecture of the group of normal cells that were gated, and displayed histograms of fluorescence at 0.5 h and 16 h of CSNP complex in which the change was negligible (Figure 7(a.2)); this is because, in normal cells (MRC5 cell line), the expression of the TfR1 is very low. The cancer cell line (H69AR) displayed a different gated cell population in the blank lecture (Figure 7(b.1)). Figure 7(b.2) displays histograms of fluorescence at 0.5 h and 16 h of the CSNP complex in which a proportional change with respect to time was observed, i.e., the longer the incubation time, the higher the level of endocytosis of Tf. Figure 7(b.3) shows a comparison between the levels of endocytosis obtained of Tf alone (positive control, black bars) and the endocytosis of the complex (grey bars); The assay results show that overexpression of the transferrin receptor TfR1, in H69AR cells is functional and facilitate endocytosis of transferrin-functionalized core-shell nanoparticles.

## 4. Discussion

Cancer is a multifactorial disease that involves the deregulation of multiple proteins which impact cell function allowing, among other things, cellular immortality in a plethora of tissues. In this sense, it is evident that the continuous search for key proteins to understand the molecular processes of the disease, as well as the search for potential biomarkers that may be targets of novel pharmacological treatments is required. We focus the attention on this last statement; one of the goals of nanomedicine is to find a protein that is highly expressed and actively participates in the molecular mechanism of the disease, thus can be blocked using different products and nanotherapeutics platforms in order to reduce its activity and cancerous phenotype [39]. Between those platforms, we can mention mCSNPs that can work through active targeting mode carrying drugs or antibodies that negatively affect the activity of the target molecule; that way we performed relative quantification based on MS in an enriched membrane of cancer cell type searching potential targets for mCSNPs. More than four hundred proteins were reported as deregulated and multiple biological pathways were altered according to the Reactome database. 

Cancer has the ability to activate the innate and acquired immune system [25,40], and this biological pathway is well represented in our results (FDR = 2.04 × 10^−11^). Important proteins involved in its regulation were found dysregulated, for example, interferon λ1 (IFNL1), which activates the expression of interferon-stimulated genes through the Jak-STAT cascade with the goal of activating immunity and cytotoxicity [41,42,43] was detected only in the MRC5 cell line, indicating an impaired defense mechanism in the H69AR cell line since these cells cannot send external signals to be eliminated.

An important characteristic of cancer cells is acquired antigenicity, which is recognized by the immune system as no self [40], whereby the ubiquitin-proteasome pathway is active to present antigens to effectors like T-cells (CD8^+^ CTLs) [44]; in this sense, the up-regulated presence of subunits, either proteasomes (PSMC3, PSMD11, and PSMD12) or immunoproteasomes (PSME1) [45], as well as enzymes associated with ubiquitination (UBA1, UBE2K, and UBR), indicates that the degradative process of antigens is active; but on the contrary, we found TAP1 protein only in MRC5 cells. This protein is very important for the binding of antigen to the major histocompatibility complex (MHC) class-I, so the results suggest that although antigens are generated, they are not bound to MHC class-I in the endoplasmic reticulum to reach the cell membrane resulting in an aberrant antigen presentation in the H69AR cell line, which is a mechanism to evade the immune response [46]. This result extends previous observations. It has been observed that in a SCLC cell line, the presence of a defective allele in TAP1 causes a change of amino acids, which leads to a defective presentation of antigen [47]; in addition, the restoration of TAP1 activity in tumor cells increases the susceptibility to CTL-mediated killing [44].

The cell cycle is a process in which the participating molecules must be perfectly coordinated to maintain the integrity of the DNA; cycle dysregulations may be due to the change in the expression of scaffolding molecules such as 14-3-3 proteins [48]. Some of these proteins were found to be up-regulated (YWHAE, YWHAG, YWHAH, and YWHAZ) within the cell cycle pathway (FDR = 8.02 × 10^−5^), and their function is to bind their substrates preferably via phosphorylation motifs (RSXpSXP or RXXXpSXP) [49]; many of the substrates like cyclin-dependent Kinases (CDK’s), p27, p53, cell division cycle proteins participate in the progression of the cell cycle so they can arrest or activate the G1-S and G2-M phases [48]. 14-3-3 proteins also are related to the assembling of microtubules because they can adapt phosphorylated Tau-protein [50], which is responsible for assembling, and form a tripartite complex (14-3-3, Tau-protein, tubulins) so it makes sense that we have detected some up-regulated tubulin proteoforms (TUBA1B, TUBAL3, TUBB, TUBB2A, and TUBB4B), and as it is widely known that microtubules are essential during mitotic progression in the cell cycle [51], many drugs like vincristine, vinblastine, and colchicine, which destabilize the microtubules, and others that stabilize them, like paclitaxel and epothilone, are used as antitumor agents [51].

On the other hand, cytoskeleton proteins like tubulin and actin as well as 14-3-3 proteins have an extremely important role during vesicle-mediated-transport (FDR = 1.51 × 10^−9^) because microtubules might be useful as a track helping to coat complex protein II (COPII), which arrives from the endoplasmic reticulum to *cis*-Golgi carrying its cargo [25]. We detected dysregulated proteins associated with COPII as SEC23A, a component of the complex [52], and SEC22B, which is a cargo that is binding with COPII through a structural epitope that works as a transport signal [53]. This protein is used as a COPII marker [54]. Additionally, we detected dysregulated components of COPI (COPB2 and COPG1) and ADP-ribosylation factor 5 (ARF5), which have the particularity that they can form COPI vesicles in the same way as ARF1 does [55], glimpsing aberrant vesicle trafficking from *cis*-Golgi to *trans*-Golgi [25,56]. Interestingly, we detected the principal components (CLTC and CLTCL1) of the third large complex with a cage structure, clathrin complex; nevertheless, they are not dysregulated. Clathrin cages use actin filaments during the endocytosis to spatially organize the endocytic machinery, deform and invaginate the plasma membrane, dissolve the cortical microfilaments barrier, generate force during or after the fission reaction, and move the vesicle into the cytoplasm [57]; in this sense, dysregulation of actin filaments could lead to endocytosis malfunction. Fascinatingly, we detected beta-actin proteins (ACTB), actin-related proteins (ACTR2, ACTR3, and ARPC3), and even actin-associated proteins [58,59] (ANK3, MYO1C, and MYH9) being mainly down-regulated, but in contrast, transferrin receptor protein 1 (TfR1) was found to be up-regulated. TfR1 has been widely studied and it could be internalized by clathrin-coated vesicles [60,61]; usually, the interaction with actin during the start of the formation of vesicles is mediated by ERM family proteins containing PDZ domains; nevertheless, the initiation of TFR1-mediated vesicle formation, which is deficient in PDZ domains, also does not use ERM proteins and consequently, actin is not involved [60], which could explain the levels of abundance reported in this work for the proteins participating in the formation of clathrin-coated vesicles. The expression levels of COPI, COPII, and clathrin-coated vesicles play a role in carcinogenic processes [62,63,64] since their up-regulation is mainly correlated with cell proliferation; nevertheless, we detected subunits of these complexes being mainly down-regulated and this is the reason why other molecular studies must be carried out to understand its role in the H69AR cell line.

We focus our attention on TfR1 because its effects can influence proteins of vesicular traffic that impact other important biological pathways altered in cancer. In addition, we consider TFR1 important for two additional reasons, (a) one of the goals of this work was to detect plasmatic membrane proteins that function as receptors that can serve as targets for functionalized mCSNPs; in this sense, TfR1 works like an important receptor expressed in membranes since transferrin-mediated iron uptake represents the major mechanism used by vertebrate cells to acquire iron from the environment [56,65]. The mechanism is based on the internalization of Fe^3+^ through its ligand [26,54], transferrin (TF). When TF binds to TfR1, the complex TF/TfR1 is endocyted by clathrin-coated pits [26,60], and inside endosomes, Fe^3+^ is converted to Fe^2+^ by metalloreductases and directed to the cytosol by its corresponding transporters [26,66]. It is important to mention that transferrin releases iron in an acidic environment [67], and, this acidic environment also is extremely useful since the potential chemotherapeutic drugs bonded to the carrying nanoparticle could be broken to make the drug bioavailable. (b) It is desirable that selected receptors have a higher expression in membranes of cell line H69AR, and as we mentioned above, TFR1 satisfies that characteristic, because there is much more abundance in tumor cells compared with non-tumor cells.

Transferrin receptor 1 overexpression has been previously described in different types of cancer, including esophageal, colon, ovarian, lung, liver, glioma, and breast cancer; and generally, an increased expression correlates with poor prognosis. Additionally, Fei Chen et al. (2021) [68], using an integrated multi-analysis of different databases, establish that the increase in TfR1 expression is present in multiple different types of cancer, in coincidence with previous studies. They also highlight that in some types of cancer such as leukemia, lung, and sarcoma, the TfR1 receptor is underexpressed.

The well-known over-expression of TfR1 indicates an important role in molecular mechanisms of cancer [69] and, despite the vast amount of information about the role of a receptor, the molecular stoichiometry of the protein between a cancerous and a normal phenotype is usually missed; although it is true that a possible target must be expressed mainly in the cancer cell, a measure of specific levels of our potential target and comparing with other cells, can give us the security to select a moiety that can discriminate between targeted and non-targeted cells to deliver therapeutic agents to targeted cells or tissues [70], therefore, we think that is important since better strategies can be designed if we know either relative or absolute quantification by proteomics methods; in this manner, TfR1 represented a good election due its exacerbated deregulation between cell lines. Additionally, the fact that its propagation is throughout the plasmatic membrane, and is not predominantly in lysosomes, exosomes, or vesicles increases the probability of a better internalization of synthetic nanotherapeutic platforms [71] like, in our case, mCSNP’s.

Some strategies regarding nanomedicine have been applied in different cell types either in vivo or in vitro, with a focus on blocking the mechanism of the action of TfR1, ranging from low molecular weight ligands like curcumin [72,73] monoclonal antibodies [74,75,76,77], miRNAs like miR-320 [78], liposomal nanoparticles targeted by a single-chain antibody fragment to the TfR1 delivering SGT-53 in combination with gemcitabine [79] and T7 and ^D^A7R dual peptides-modified liposomes codelivery doxorubicin and vincristine [80,81]. Nanoparticles synthesized with Poly (lactic-co-glycolic acid) (PLGA) carrying surfactant 77KS and doxorubicin using transferrin as ligand targeting on TfR1, have also been tested with positive effects [82], but no attempt has yet been made to use transferrin functionalized mCSNPs, which could serve as a promising strategy since our results indicate that this can be applied but, this nanoparticle must codelivery some anticancer drug in a specific and controlled manner [83,84,85], exploiting endocytosis and allowing for better cellular and subcellular targeting of drugs, leading to therapies with better efficacy and improved tolerability in order to diminish the cancerous phenotype; which is our next objective. Although this approach is conceptually straightforward and has been successfully used by marketed nanomedicines, it is often challenging practically [86,87].

Quantitative proteomics studies and the use of core-shell nanoparticles, which we have previously shown to have magnetic properties and are attractive for application in the fields of theranostic imaging, cell tracking, hyperthermia, and drug delivery [20], provided an opportune tool to functionalize these nanoparticles with transferrin and target the TfR1 receptor.

## 5. Conclusions

The use of label-free DIA mass spectrometry in combination with ion-mobility (IM-MS) allowed the identification of TfR1, which was found significantly overexpressed in lung tumor cells. TFfR1 is 32 times more abundant in tumors than in normal cells, which could be used to cover the mCSNP’s surface together with drugs or antibodies for SCLC treatment. This approach would be valuable for the therapeutic effect, to have a more significant impact on the tumor cells, and, to some extent, to decrease the damage to healthy cells. With the use of magnetic Au-CoFe_2_O_4_ core-shell nanoparticles with paramagnetic properties, the main advantage of this nanoparticle nucleus-envelope is that it is constituted of two different nanomaterials, which can be effectively used for treatment. The gold coating provides a versatile functional surface platform as well as improves bioavailability and enhances liver and kidney clearance metabolism, capable of functionalization with transferrin, and represents a new alternative for early diagnosis and targeted and controlled drug delivery, for lung cancer and other types of cancer.

## Data Availability

The mass spectrometry proteomics data have been deposited to the ProteomeXchange Consortium via the PRIDE partner repository with the dataset identifier PXD015405.

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
