# Peer review of "Identification of Transferrin Receptor 1 (TfR1) Overexpressed in Lung Cancer Cells, and Internalization of Magnetic Au-CoFe2O4 Core-Shell Nanoparticles Functionalized with Its Ligand in a Cellular Model of Small Cell Lung Cancer (SCLC)"

_pharmaceutics, 2022, doi:10.3390/pharmaceutics14081715_

Round 1

Reviewer 1 Report

The authors compared the membrane protein content between a small cell lung cancer (SCLC) cell line H69AR and the non-cancerous lung fibroblast MRC5 by label-Free mass spectrometry in combination with an ion-mobility approach. They identified overexpressed proteins in the cancerous cell line and they focused on the transferrin receptor (TFRC), as a marker for lung cancer. Then, they functionalized the magnetic core-shell nanoparticles (mCSNP) with transferrin, which is the target molecule for binding to the TFRC. The kinetics of endocytosis was then evaluated by flow cytometry methodology demonstrating that the functionalized mCSNP were internalized.

The title could be more precise and include TFRC as the overexpressed receptor. Several figures and legends should be revised. Here are different remarks:

CD71 should be clearly identified as the transferrin receptor in the main text

Lane 45-46 : Revision of the sentence comparing number of death  in 2020 with a % in 2018.

Lane 60 : non instead of No small cell lung cancer (NSCLC)

Lane 120 “A normal lung cell line MRC5 (ATCC® CCL171)…” MRC5 are lung fibroblast but not a normal epithelial lung cell.

Lane 199: Why adding 0.5% FBS in sample 1?

Lane 207 Transferrin Alexa “fluor” replaces “flour” and add “(TF-AF488)”

Lane 279 T3.1 to be replaced by 3.1

Lane 168 what is FDR?

Figure 1 : legend Biological pathways are fuzzy and hard to read.

Membrane protein extraction: the authors could explain what types of membranes are extracted with the Mem-PER plus membrane Protein extraction kit.

Sequencing shows different types of proteins, not all of them are membrane protein (hNRP, helicases, eIF,….). The authors should discuss this result.

Figure 3 A has to be redone. It is hard to read and understand.

Figure 3 legend Lane 359 This result is in line 359 with the literature reporting that CD71 (a homodimeric type II membrane glycoprotein, 360 95 kDa) binds to and assists entry of its ligand into cells for the delivery of iron10. This sentence has to be included in the main text and not in the legend.

Figure 3 C1 legend should be improved to understand B++, B+-; B--; B-+.

Figure 4: How can it be proved from the picture that “TFRC was distributed throughout the plasmatic membrane”. The green labeling seems to be all over the cytoplasm. What kind of labelling was obtained with MRC5 cells?

Figure 6 legend should be improved. A? B? Y-axis labelling?

Figure 7a.1 why a MRC5 cell population (Bottom left) is excluded from the cytometry fluorescence analysis?

Figure 7 a.2 where is the blank panel? (Lane 460).

Lane 457 Figure 7 legend: No result at t= 3h are shown.

Figure 7 b.3 How to explain the high variability in the endocytosis of CSNPs-Tfn compared to Tfn? The results are presented as the mean +/- SE?

Figure 7 : abbreviation TF should be used in the graph legend b.3 instead of Tnf

Figure 7 : what are the letters A/C/D/F on the different graphs of the figure?

Lanes 413-433: The syntax of the paragraph should be improved. Lane 423 is the beginning of one sentence of 9 lanes.

Lane 431 : How the quantification of 5 µg of protein bound to the nanoparticle was obtained ?

Lane 447 “this means that the TFRC is mostly expressed and active on the cancer cells than in 447 normal cells.” The syntax should be revised.

The different biological pathways identified through the TFRC proteins interactome are discussed from lane 482. Part of this discussion could be included in the result part as a description of the figure 2.

The discussion should be shortened and should emphasized on the originality and the interest of the magnetic Au-CoFe2O4 core-shell nanoparticles. The transferrin receptor is already a well-known target highly expressed in cancerous cells including lung cells.

Reviewer 2 Report

This is an excellent attempt in using new magnetic technolgy of nanomedicine for cancer diagnosis and treatment. A few points  should be included

1. The authors mentined that CoFerricoxide is toxic and they need to coat it with Au. They should explain why they did not use the safer Ferric oxide nanoparticles

2. The authors gave 82 refences yet they only include about 5 references on magnetic nanoparticles and nanomedicine. This is a very large area but they should at least include a recent 2019 general review of the whole area from Artificial Cells, Nanomedicine,Biotechnology 47: 997-1013,2019

3. Method: In the preparation of the nanoparticles, they should give the exact amounts of each component thay use for the preparation

4. The figures for the nanoparticles indeed show very uniform particles. However, they appear to be all aggregated together, They should explain whether this is agregaation due to the preparative procedure for taking the TEM. I do not think the aggregate could be injected

Fig 7 b3  What is the stastical analysis for significance of the differences. There seems to be quiet some SD of the results.

Reviewer 3 Report

This referee think  the article is very clear and exhaustive. It describes the scientific work done in a very simple but effective way, so that also people not very familiar with the argument can  easily understand.

Reviewer 4 Report

In the manuscript titled " Identification of an overexpressed receptor in lung cancer cells, and internalization of magnetic Au-CoFe2O4 core-shell nanoparticles functionalized with its ligand in a cellular model of small cell lung cancer (SCLC).", Villalobos-Manzo et al. used label-free mass spectrometry in combination with an ion-mobility (IM-MS) approach to identify a marker for lung cancer, and functionalized magnetic core-shell nanoparticles with its ligand to improve the affinity for small cell lung cancer cells. This work is well developed and supported by the results, and the authors did adequate data analysis of quantitative proteomics studies. After addressing the following questions properly, the manuscript could be considered for publication.

1. The authors are suggested to move the figure caption of Figure 2 under the figure, not above the figure.

2. The authors are suggested to replace Figures 1 and 3 with new figures that have a higher resolution (every word should be clear enough for reading).

3. L446 makes me confused, I cannot find any blue bars and red bars in Figure 7.b.3. Please correct it.

4. The authors are suggested to add a paragraph to discuss why the error bars of the endocytosis of the complex were much bigger than the positive control, and why the endocytosis of the complex was higher than the positive control at 0.5 h but was lower than the positive control at 8 and 16 h (Figure 7.b.3).

5. The introduction should contain more dated literature. In its current form, no article is from 2019 or newer. If no one else has been working on that topic in recent years, one may wonder if the impact of that research is actually high.

Reviewer 5 Report

The article titled “Identification of an overexpressed receptor in lung cancer cells, and internalization of magnetic Au-CoFe2O4 core-shell nanoparticles functionalized with its ligand in a cellular model of small cell lung cancer (SCLC) by Villalobos-Manzo et al. has been reviewed. Following are some of my comments on this work.

11.       The first sentence of the Introduction has some data which is informative but since one is number while the other is a percentage, the information is not so clear.

22.       Line Nos. 81-83: The number ~12 times is not so convincing, instead the information that the number of vomiting is high could be presented in a better way.

33.       In materials ad methods, the units are to be written following the standard way, e.g., mL instead of ml, etc. Same is for rest of the manuscript.

44.       Line No. 226: What does this ‘TD’ represent?

55.       Line 234: RPM to convert to gyration.

66.       Mention the power (if available) of the sonicator used.

77.       What is the reason of choosing MUA for the surface modification?

88.       Provide a schematic presentation to present the synthetic procedure of the nanoparticles and their surface modifications.

99.       Line No. 279: ‘T3.1. Proteomic Analysis’, typographical mistake!

110.   Table S1 can be included in the main supplementary file.

111.   The characterization of the synthesized nanoparticles is to be done well. Although the authors have claimed the dimension of the nanoparticles as 5 – 10 nm, it could not be verified well through the TEM image presented in Figure 5. The image is nor clear either. Kindly provide a clear image and an image that would be of smaller scale (say ~ 20 nm) along with the image of 100 nm scale. So, the size and its distribution both can be verified. Also, the XRD data is missing. Figure 1 of the paper (DOI: 10.1039/d2tb00246a) can be a guide to carry out these properly. There are many more such papers that can be consulted for this.

112.   It is known that the coating of NPs by gold improves cell viability, and the authors have also mentioned it in the manuscript. However, authors have not carried out any experiment to show how much improvement it has caused in their work. Cytotoxicity investigation of the synthesized materials cannot be ignored.

113.   In Figure 7, can a time-course assay be included to show the endocytosis progress?

114.   What is the purpose of including magnetic property to the NPs for this work?

115.   Finally, English correction is required in various places of the manuscript to convey the message more clearly.

Round 2

Reviewer 1 Report

I appreciate the response of the authors to my different requests. Here are mainly syntax remarks:

P3 lane 120 TrF1 instead of TfR1

P3 Lane 137 “and adding a  protease inhibitor cocktail” may not be at the right place

P5 lane 211 “ mouse anti-human FOLRI AF488-conjugated anti-mouse antibody” this sentence has to be checked

P5 lane 218 “and” to be erased

P8 lane 342 “because ? impacts » the subject is missing.

P10 lane 404. The last sentence of the paragraph articulated by “Additionally” is not clear. What is link between the cell model and the lumens of the vesicles ?

P11 lane 441: close the parenthesis

P11 lane 446 “his” replaced by its.

P13 lane 493 “an 11-carbon linker (MUA),(point?) the binding of transferrin” : The binding of transferin should be a new sentence.

P13 lane 497 the transferrin band can be observed, it is a 79 kDa glycoprotein, of which the quantitative analysis of this band was performed … : the syntax of this sentence has to be checked

P14 fig 7 b.3 The expression of the data as the mean +/-SEM should be mentioned in the legend text.  The authors maintained the letters on the cytometry graphs. No reference is made to these letters in the legend. Maybe, the letters could be deleted.

Reviewer 5 Report

The authors have successfully responded to the comments, however a few of them are not satisfactory.

Earlier Reviewer’s Comment: Line Nos. 81-83: The number ~12 times is not so convincing, instead the information that the number of vomiting is high could be presented in a better way.

Authors’ Reply: We agreed with the observation,

The correction was made by removing that information that might not be the principal idea of that line.

Comment 1: Though the authors mentioned that the correction is made by removing the information, the manuscript has the same information remaining (Line Nos. 86-88). Modifying this sentence would be better.

Comment 2: The authors have prepared the Scheme 1 very well which conveys the message clearly, it is praiseworthy. The scheme captain needs to have reference of the Springer Protocols, so that readers can find it easily.

Comment 3: Although the authors have provided the TEM image of the nanoparticles in scale 20 nm, the XRD data is not yet provided. If the authors think it is not required, then provide the reason.
